# The Renin–Angiotensin System in the Tumor Microenvironment of Glioblastoma

**DOI:** 10.3390/cancers13164004

**Published:** 2021-08-09

**Authors:** Michael O’Rawe, Ethan J. Kilmister, Theo Mantamadiotis, Andrew H. Kaye, Swee T. Tan, Agadha C. Wickremesekera

**Affiliations:** 1Department of Neurosurgery, Wellington Regional Hospital, Wellington 6021, New Zealand; michael.orawe@ccdhb.org.nz; 2Gillies McIndoe Research Institute, Wellington 6021, New Zealand; ethankilmister467@gmail.com; 3Department of Surgery, The Royal Melbourne Hospital, The University of Melbourne, Parkville, VIC 3050, Australia; theo.mantamadiotis@unimelb.edu.au; 4Department of Neurosurgery, Hadassah Hebrew University Medical Centre, Jerusalem 91120, Israel; andrewk@hadassah.org.il; 5Wellington Regional Plastic, Maxillofacial & Burns Unit, Hutt Hospital, Lower Hutt 5040, New Zealand

**Keywords:** glioblastoma, renin–angiotensin system, pluripotent stem cells, organoids, cancer stem cells, cancer stem cell niche, tumor microenvironment

## Abstract

**Simple Summary:**

Glioblastoma (GB) is the most aggressive brain cancer in humans. Patient survival outcomes have remained dismal despite intensive research over the past 50 years, with a median overall survival of only 14.6 months. We highlight the critical role of the renin–angiotensin system (RAS) on GB cancer stem cells and the tumor microenvironment which, in turn, influences cancer stem cells in driving tumorigenesis and treatment resistance. We present recent developments and underscore the need for further research into the GB tumor microenvironment. We discuss the novel therapeutic targeting of the RAS using existing commonly available medications and utilizing model systems to further this critical investigation.

**Abstract:**

Glioblastoma (GB) is an aggressive primary brain tumor. Despite intensive research over the past 50 years, little advance has been made to improve the poor outcome, with an overall median survival of 14.6 months following standard treatment. Local recurrence is inevitable due to the quiescent cancer stem cells (CSCs) in GB that co-express stemness-associated markers and components of the renin–angiotensin system (RAS). The dynamic and heterogeneous tumor microenvironment (TME) plays a fundamental role in tumor development, progression, invasiveness, and therapy resistance. There is increasing evidence showing the critical role of the RAS in the TME influencing CSCs via its upstream and downstream pathways. Drugs that alter the hallmarks of cancer by modulating the RAS present a potential new therapeutic alternative or adjunct to conventional treatment of GB. Cerebral and GB organoids may offer a cost-effective method for evaluating the efficacy of RAS-modulating drugs on GB. We review the nexus between the GB TME, CSC niche, and the RAS, and propose re-purposed RAS-modulating drugs as a potential therapeutic alternative or adjunct to current standard therapy for GB.

## 1. Introduction

Glioblastoma (GB), the most common and most aggressive primary brain cancer in humans, is classified as a WHO grade IV astrocytoma, and is characterized by microvascular proliferation and central necrosis [1]. Primary GB arises de novo and accounts for 90% of cases with a predilection for older individuals, while secondary GB arises from low-grade astrocytoma and affects younger patients [2]. GB has been categorized into four distinct molecular subtypes: classical, mesenchymal, neural, and proneural [3], although other studies have only identified classical, mesenchymal, and proneural subtypes [4]. The classical subtype includes amplification or mutation of epidermal growth factor receptor (EGFR), the mesenchymal subtype includes deletions of the 17q11.2 region containing the gene *NF1*, and the proneural subtype is characterized by high levels of platelet-derived growth factor receptor α (PDGFRα) expression and point mutations in *isocitrate dehydrogenase 1* (*IDH1*) and *p53* [3].

Various genetic or epigenetic changes may affect the prognosis of GB patients including *IDH* mutations and O6-methylguanine-DNA methyltransferase (MGMT) methylation status. GB may be divided into IDH-wild-type and IDH-mutant tumors. IDH is an enzyme that catalyzes oxidative decarboxylation of isocitrate to 2-oxoglutarate. The most common mutation in GB affects *IDH1* with a single amino acid missense mutation at arginine 132 which is replaced by histidine [5]. IDH-wild-type GB is more common, tends to arise de novo, and is generally more aggressive with a worse prognosis than IDH-mutant GB. By contrast, IDH-mutant GB is predominantly observed in secondary GB and is associated with a better prognosis [6]. The current standard treatment for GB involves maximal safe surgical resection with adjuvant chemotherapy and radiotherapy, known as the Stupp protocol [7]. Temozolomide, an alkylating agent, is used as first-line chemotherapy for GB with its efficacy related to the methylation status of the *MGMT* promoter [8]. *MGMT* methylation is associated with an improved overall survival in GB patients [9]. Despite this intensive treatment, tumor recurrence in GB patients is inevitable with an overall median survival time of 14.6 months with a range of 12–14 months which has not changed since the introduction of the Stupp protocol in 2005 [10,11].

We reviewed the dynamic relationship between the tumor microenvironment (TME), the RAS, and cancer stem cells (CSCs) in GB. We speculate that RAS-modulating drugs may offer a potential therapeutic alternative or adjunct to current standard therapy. Further functional and epidemiological studies are required to investigate the efficacy of RAS-targeting drugs in the treatment of GB.

## 2. GB Tumor Microenvironment

The GB tumor microenvironment (TME) is highly heterogeneous and consists of cancer cells and non-cancer cells. Non-cancer cell types include immune cells, such as tumor-associated macrophages (TAMs), resident glial cells, peripheral macrophages, endothelial cells, pericytes, astrocytes, CSCs, fibroblasts, and other components such as the extracellular matrix (ECM) [12]. Given the rarity of extracranial metastasis from GB [13], it appears that GB development requires the unique intracerebral microenvironment inclusive of the blood–brain barrier (BBB) [14]. The TME, with emphasis on glioma-associated microglia/macrophages, pericytes, and reactive astrocytes, is increasingly recognized to play a critical role in GB development and progression [15]. The idea that cytokines, growth factors, chemokines, inflammatory mediators, and remodeling enzymes are involved in intra- and inter-cellular communications within the TME is not novel [16]. Additionally, constant communication between GB cells and the surrounding TME [14] is facilitated by extracellular vesicles that expedite bi-directional cross-talk within the TME [12,17].

Anatomically distinct regions of the TME, known as tumor niches, are thought to contain CSCs and play a fundamental role in the regulation of metabolism, immune surveillance, survival, invasion, and self-maintenance with the renin–angiotensin system (RAS) playing a critical role [15,18,19]. The GB TME may consist of several distinct tumor niches including the hypoxic tumor niche, the perivascular or angiogenic tumor niche, and the vascular-invasive tumor niche. The perivascular niche contains CSCs in close juxtaposition with the abnormal angiogenic vasculature and provides a supportive environment for CSC growth, maintenance, and survival. The vasculature in the hypoxic tumor niche is either non-functional or has regressed, leading to areas of necrosis that are surrounded by rows of hypoxic palisading tumor cells [20]. The vascular-invasive tumor niche contains tumor cells co-opted with normal blood vessels that migrate deep into the brain parenchyma [20]. 

GB is highly vascular and is characterized by extensive neovascularization and pathological angiogenesis predominantly induced by vascular endothelial growth factor (VEGF), which is produced by tumor cells, CSCs, and immune cells [21,22]. Other angiogenic factors, such as transforming growth factor-β_1_ (TGF-β_1_), platelet-derived growth factor-BB, and fibroblast growth factor-2, may also play a role in the pathological angiogenesis [23,24]. In addition to endothelial proliferation, bone marrow-derived endothelial and pericyte progenitor cells may be recruited and incorporated into the growing vessels [25]. There is also evidence that CSCs may be involved in neovascularization by differentiating into endothelial cells or pericytes in GB [26,27,28]. Increased VEGF expression also fosters an immunosuppressive microenvironment that enables tumors, including GB, to evade host immune surveillance [29]. The abnormal vasculature in GB includes dilated and leaky vessels and glomeruloid microvascular proliferation in which endothelial cells and pericytes form poorly organized vascular structures, which effectively disrupt the BBB, leading to cerebral edema. In addition, the blood–brain tumor barrier (BBTB) hinders drug delivery to the tumor [30].

The BBB is a highly specialized, selectively permeable barrier between the brain and the systemic blood supply that helps to maintain homeostasis of the cerebral microenvironment. The structure of the BBB includes endothelial cells with tight junctions, adherens junctions, astrocytes, pericytes, and the basement membrane [31]. The BBB plays several fundamental roles, including supplying the brain with essential nutrients, such as oxygen and glucose, mediating the efflux of waste products, facilitating the movement of nutrients and plasma proteins, and restricting toxins into the central nervous system (CNS) [32]. Disruption of the BBB and its tight regulation of the cerebral microenvironment leads to increased blood vessel permeability with plasma and fluid leakage into the tumor tissue causing cerebral edema and raised interstitial and intracranial pressure [33]. The combination of abnormal vasculature in GB and the disruption of the BBB leads to impaired blood flow and reduced oxygen delivery within the tumor [34]. Microvascular thrombosis may also occur causing occlusion of the blood vessels, further promoting intra-tumoral hypoxia, leading to pseudo-palisading necrosis [35]. Hypoxia is also a consequence of increased oxygen diffusion distance due to the fact of tumor growth and expansion [34], which may, in and of itself, be a key regulator of tumor cell survival, stemness, and immune surveillance in the TME [36,37,38]. Hypoxia also sustains tumor cell proliferation, invasiveness, and contributes to chemotherapy and radiotherapy resistance. This occurs via inhibition of free radicals, which reduces the efficacy of radiotherapy [39], and through upregulation of the multi-drug resistance gene, *MDR1/ABCB1*, which reduces chemotherapy effectiveness. Hypoxia-inducible factor-1 (HIF-1) and HIF-2 mediate the response to hypoxia on a molecular level in GB [40] and may potentially modify CSCs [41]. The GB microenvironmental niche also consists of pseudo-palisading glioma cells that upregulate HIF proteins, inducing expression of factors, such as VEGF and interleukin 8 (IL-8), which are implicated in tumor cell survival, metabolism, invasion, and angiogenesis. The resultant cross-talk releases pro-inflammatory signals from the areas of necrosis in the hypoxic tumor niche into the surrounding TME, promoting immunosuppression, and angiogenesis [42].

Immune cells, including circulating monocytes, neutrophils, and myeloid-derived suppressor cells (MDSCs), are another source of angiogenic factors. In ovarian cancer, MDSCs increase CSC characteristics by increasing microRNA-101 expression, which induces the expression of stemness genes [43]. It is interesting to speculate that MDSCs also regulate the stemness of CSCs within the GB TME via this mechanism (Figure 1). These cells may enter the brain as a result of breakdown of the BBB in GB and the production of tumor-derived chemokines and cytokines, contributing to the immunosuppressive GB TME [44,45,46]. TAMs are the dominant immune cell population in GB and may include resident microglial cells and peripheral macrophages [47,48]. Traditionally, TAMs have been defined as either anti-tumoral M1/Th1 (classical-activated macrophages) or pro-tumoral M2/Th2 (alternative-activated macrophages) phenotypes. M1 macrophages foster the inflammatory response by secreting pro-inflammatory cytokines such as IL-12, tumor necrosis factor-α (TNF-α), CXCL-10, and interferon-γ (IFN-γ) and produce high levels of nitric oxide synthase to exert anti-tumor cell activity (Figure 1). M2 macrophages, on the other hand, play a key immunosuppressive function by secreting anti-inflammatory cytokines, such as IL-10, IL-13, and IL-4, and express abundant arginase-1, mannose receptor CD206, and scavenger receptors to promote tumor progression [49,50,51]. The release of TGF-β by TAMs has been shown to induce matrix metalloproteinase 9 (MMP9) and, thus, increase CSC invasiveness [52]. A more recent study has demonstrated that the TAM population is in a constant state of transition or plasticity between the two phenotypes and that M1 phenotype expression may be enhanced by TME changes or therapeutic interventions [51]. Resident microglia are present within the brain, but it is the recruitment of peripheral macrophages to the GB TAM pool, in particular, that may mediate tumor phagocytosis with disruption of the signal regulatory protein α receptor (SIRP-α)–CD47 axis. This facilitates immune evasion because the antiphagocytic “don’t eat me” surface protein CD47 is upregulated, which binds to SIRP-α on phagocytic cells to inhibit phagocytosis [53]. However, even in the absence of macrophages, resident microglia may be transformed into effector cells of tumor cell phagocytosis, in response to anti-CD47 blockade [54]. In models of pancreatic ductal adenocarcinoma, for example, RP-182 may selectively induce conformational switching of the mannose receptor CD206, which is expressed on the M2 TAM phenotype, ultimately reprogramming M2-like TAMs into an anti-tumor M1-like phenotype [55]. The immunosuppressive phenotype of TAMs may be controlled by long-chain fatty acid metabolism, and chemical inhibitors targeting this metabolic pathway may block TAM polarization in vitro and tumor growth in vivo [56]. GB-derived exosomes may reprogram M1 macrophages to M2 macrophages and condition M2 macrophages to become strongly immunosuppressive TAMs [57].

## 3. Glioblastoma Cancer Stem Cells

The CSC concept proposes that a small distinct population of cells within a tumor with self-renewal capability are responsible for driving tumorigenesis [58,59]. These CSCs may be defined as stem cell-like cells within a tumor that also have the capacity for proliferation and multi-potency. This may be regarded as a functional definition insofar as CSCs may be characterized through the generation of serially transplantable tumors that faithfully recapitulate the parent tumor [60]. There is marked intra- and inter-tumoral heterogeneity including, differing numbers of highly tumorigenic CSCs [61]. Such heterogeneity may be best explained by a combination of different models of cancer, including the stochastic model (also known as the clonal evolution model), the CSC concept of cancer (also known as the hierarchical model of cancer), and the concept of plasticity [62,63]. 

The traditional model of cancer is predicated on the stochastic model of carcinogenesis which proposes that cancer cells are derived from normal cells that acquire genetic and/or epigenetic mutations resulting in typically unidirectional transitions from benign to malignant cells. These malignant tumor cells have unrestricted division capacities and their high mutation rates increase the likelihood of successive generations of cloned cells being adapted to the selection pressures of the tumor site. However, the stochastic model does not fully account for all aspects of cancer biology including tumor recurrence following treatment [64].

In contrast, the CSC concept of cancer proposes that CSCs contribute to carcinogenesis, invasion, metastasis, therapy resistance, and recurrence [65,66]. CSCs divide asymmetrically into non-tumorigenic cancer cells, which form the bulk of a tumor, and identical highly tumorigenic but less abundant CSCs, which sit at the apex of the cellular hierarchy [67]. CSCs have been postulated to originate from non-malignant stem cells or progenitor cells [66] or dedifferentiated cancer cells [68]. The overlap between the stochastic model and the CSC concept may be explained by the concept of cellular plasticity whereby cancer cells may reversibly transition between stem-like and non-stem-like cell states [69]. This process of transition may be driven by embryonic stem cell (ESC)-associated regulatory networks and may be affected by the dynamic TME including the CSC niche [70]. Moreover, certain cancer cells may de-differentiate and re-enter the CSC pool, thus regaining the capacity for tumorigenesis and clonal expansion [71].

CSCs have been found in many different cancer types, including myeloid leukemia [72], pancreatic cancer [73], breast cancer [74], oral cavity squamous cell carcinoma (SCC) [75,76,77], primary [78] and metastatic [79] cutaneous SCC, primary [80] and metastatic [81] colon adenocarcinoma, metastatic malignant melanoma [82,83], and GB [84]. The aggressive nature of GB and its resistance to conventional therapy has been attributed to the presence of CSCs [85] that were first postulated in human brain tumors, identified by their expression of the neural stem cell surface marker CD133 [86]. Stem-like neural precursor cells responsible for the growth and recurrence in serial transplantations were identified in GB [87]. The presence of such quiescent CSCs is well-supported in the literature and the interaction of such cells with the ECM and TME factors, including TGF-β and hypoxia, may contribute to their resistance to conventional therapy [88] (Figure 1). There is evidence that CSCs may be stimulated to differentiate into endothelial cells by activating Notch1 signaling [89] and may be associated with induction of cytokines, MMPs, and adhesion proteins in the TME [90].

A crucial function of stem cells is self-renewal, for which the Notch, Sonic hedgehog, and Wnt signaling pathways may be essential [91] (Figure 1). GB expresses a number of stemness-associated markers including cell surface markers (CD133, CD15, A2B5, and L1CAM), cytoskeletal proteins (nestin), transcription factors (SOX2, NANOG, and OCT4), post-transcriptional factors (Musashi1), and polycomb transcriptional suppressors (Bmi1 and Ezh2) [85]. There is also evidence of plasticity and bi-directional interconversion between CSCs and cancer cells [92]. In a landmark study, pluripotent stem cells were formed from reprogrammed mouse embryonic and adult fibroblasts by the addition of transcription factors OCT4, SOX2, c-MYC, and KLF4 [93]. These factors, in addition to NANOG, which are expressed by ESCs, have been identified in GB [84]. The capacity of GB cells for perpetual self-renewal may rely on the contribution from transcription factors such as OCT4 and SOX2 [85]. SOX2 is highly expressed in GB [84] and may play a key role in maintaining plasticity for bi-directional cellular conversion in GB [94]. Moreover, silencing of SOX2 inhibits tumor proliferation in GB [95] and, thus, it may be a potential therapeutic target in the treatment of GB [96]. Another potential therapeutic target involves the JAK–STAT3 signaling pathway which is also associated with the self-renewal capacity of GB. Inhibition of this pathway may impede the migratory and invasive potential of GB by decreasing activation of the transcription factor STAT3 and, thus, reducing the levels of MMPs and associated invadopodia activity [97]. In addition, STAT3 binding to the *Notch1* promoter inhibits this signaling pathway and may impede the maintenance of glioma stem-like cells while reducing the expression of glioma stem cell markers CD133, SOX2, and nestin [98] (Figure 1).

## 4. The Renin–Angiotensin System and Convergent Signaling Pathways in Glioblastoma

The RAS has been proposed to play an important role in the TME [19] in various cancer types, including lung cancer, through its effect on tumor cells, non-malignant cells, hypoxia, angiogenesis, and the inflammatory response [99]. The RAS is a complex physiological system and has a multitude of interactions with many different convergent signaling pathways that operate in carcinogenesis, some of which lie outside the scope of this article.

Classically, the RAS regulates blood pressure and electrolyte and fluid homeostasis involving primarily the renal, cardiovascular, and endocrine systems [100]. The RAS pathway is composed of multiple steps culminating in the formation of the main effector hormone, angiotensin II (ATII) [101]. Activity of this key homeostatic system in the CNS is well documented [102]. In this review article, RAS inhibition broadly refers to inhibition of any of the components of the RAS, reducing its downstream effects.

Angiotensinogen, primarily synthesized in the liver by hepatocytes, is cleaved by renin, to form angiotensin I (AT1) [103]. Angiotensinogen is synthesized and secreted by astrocytes and is converted to several neuroactive peptides [104,105]. Angiotensinogen is also produced within neurons, which can secrete or retain it intracellularly. These neuroactive peptides bind their respective receptors within the local microenvironment to induce receptor signaling by different cell types [104,105]. Renin is physiologically derived from the juxtaglomerular apparatus in the kidneys and its release is tightly regulated by macula densa and local baroreceptors [106]. Renin is formed by the binding of prorenin to the prorenin receptor (PRR) [107] and is also catalyzed by enzymes such as cathepsins B, D, and G [108,109,110,111]. ATI is converted to ATII by angiotensin-converting enzyme (ACE), also known as ACE1, which is primarily found in the lungs [112]. ATII binds to ATII receptor 1 (AT_1_R) and ATII receptor 2 (AT_2_R) [113]. ATII binding to AT_1_R causes MAPK–STAT3 activation [114] and phosphatidylinositol signaling, which increases cytosolic Ca^2+^ and effects mitogenesis [115]. AT_1_R signaling increases RAS activity in the TME, and the formation of NF-κB and TGF-β_1_ which promotes cellular proliferation, inflammation, and angiogenesis [116]. AT_2_R activation by ATII inhibits cellular growth and enhances apoptosis [116]. ATII can be further converted into angiotensin III (ATIII), and then angiotensin IV (ATIV) by aminopeptidase-A (AP-A) and aminopeptidase-N (AP-N), respectively. ATIV binds to ATII receptor 4 (AT_4_R), and in high concentrations, may bind to AT_1_R. Angiotensin (1–7) (Ang(1–7)) is produced by the cleavage of either ATI by neutral endopeptidase (NEP) or ATII by ACE2, an isoform of ACE. Ang(1–7) binds to Mas receptors (MasRs) [117,118]. ATI may also be cleaved by ACE2 to form Ang(1–9), which can be cleaved by ACE1 and is converted to Ang(1–7), which in addition to binding to MasRs, can also bind to AT_2_R with low affinity, and Mas-related-G protein coupled receptors (MrgDs) [119]. MrgDs are a recently discovered component of the RAS [102], and their role in the GB TME is yet to be defined. Lastly, the primary ligand for MrgDs is almandine, an Ang(1–7) analog formed by decarboxylation of Ang(1–7) [102] (Figure 2). 

Key components of the RAS are also activated in CNS diseases [101]. Renin, and its precursor prorenin, are expressed variably in neurons, astrocytes, oligodendrocytes, and microglia in different regions of the brain [120,121]. PRR is widely distributed in different organs throughout the body including the brain, eyes, and immune system [122]. ACE1 is expressed in areas of the brain involved in blood pressure control and homeostasis including the choroid plexus, organum vasculosum of the lamina terminalis, subfornical organ, and area postrema [104]. ACE2 is found in the endothelium of the brain in various regions including the cortex and brainstem [123]. ACE2 contributes to the neuroprotective ACE2/Ang(1–7)MasR signaling axis by converting ATII to Ang(1–7) which is a ligand for MasR [124].

The RAS, as a constituent of the TME, is involved in several hallmarks of cancer, including angiogenesis, hypoxia, and tumor cell proliferation [125]. Components of the RAS are expressed in different types of cancer including colon adenocarcinoma [126] and malignant melanoma [127]. RAS components are also expressed by CSCs in oral cavity SCC [128,129], renal clear cell carcinoma [130], primary [131], and metastatic [132], cutaneous SCC, metastatic colon adenocarcinoma [133], metastatic malignant melanoma [82,83], and GB [134]. In GB, PRR, AT_1_R, and AT_2_R are co-expressed with stemness-associated markers [134]. PRR is highly expressed in GB compared with lower-grade gliomas; this higher expression of PRR in higher-grade glioma is notable as the Wnt/β-catenin signaling pathway is implicated in the self-renewal of stem cells [135] (Figure 1). 

The Wnt/β-catenin signaling pathway, which sits downstream of the RAS, is implicated in tumor initiation in several cancer types [136]. In brief, this pathway results in active β-catenin translocating into the nucleus, upregulating the expression of oncogenes such as *c-Myc*, *AXIN2,* and *CCND1* [136]. PRR is a component of the Wnt receptor complex and acts as an adapter between vacuolar H^+^-adenosine triphosphate (V-ATPase) and low-density lipoprotein receptor-related protein 6. V-ATPase, a proton pump, is essential for cellular acidification and is involved in the mechanism for β-catenin activation [137]. This process facilitates binding of Wnts to their respective Wnt receptor complex [138]. Further, PRR promotes brain cancers via the Wnt/β-catenin signaling pathway, and in addition to being a membrane receptor, exists in the cytoplasm and increases the protein expression of Wnt2 within glioma cells [135]. This evidence underscores the PRR as a potential oncoprotein via Wnt/β-catenin pathway-related carcinogenesis [136], which influences cell stemness [139], tumorigenesis, and cellular proliferation [140,141]. Renin is expressed in GB and may contribute to the mechanisms of neovascularization in GB [142]. Furthermore, downregulation of the Ang(1–7)/MAS signaling axis by podocalyxin results in enhanced GB cell invasion and proliferation [143]. Finally, bypass loops of the RAS involving various cathepsins that may also contribute to the proliferative activity in GB, for example, cathepsin G coverts ATI to AII and from AGT directly to ATII, which binds to AT_1_R, to promote cancer progression [144,145,146]. GB CSCs have been shown to secrete Wnt-induced signaling protein 1 (WISP1) that promotes the survival of both the CSCs and M2 TAMs to promote a pro-TME [147] (Figure 1).

Other related signaling pathways, such as the PI3K/AKT/mammalian target of rapamycin (mTOR) and Ras/RAF/MEK/ERK pathways within the GB TME, downstream to the RAS, are activated via AT_1_R and PRR signal transduction. MAPK/ERK signaling is activated upon binding of renin or prorenin to PRR, and this upregulates ERK1/2 in various cell types including neurons [148]. ERK1/2 activation induces TGF-β_1_ formation and cellular proliferation, both of which influence cancer development [136]. Supporting this is the fact that silencing of PRR downregulates expression of ERK1/2, AKT, and NF-κB [149]. Additionally, PRR activation leads to the production of reactive oxygen species, which activates both the PI3K/AKT/mTOR and Ras/RAF/MEK/ERK pathways (Figure 2). It is interesting to speculate that both pathways operate in conjunction with the RAS and Wnt/β-catenin to influence proliferation, survival, stemness, and invasiveness of CSCs within the GB TME.

The use of RAS inhibitors (RASis) in the treatment of cancer may mitigate the cytotoxic treatment-related adverse effects experienced by cancer patients to improve their overall quality of life [150]. A meta-analysis of 17 observational studies by Shen et al. [123] show RASis are associated with a reduced risk of cancer [151]. A prospective population-based study also shows long-term (>3 years) administration of RASis is associated with a decreased risk of cancer in patients with a DD genotype, which is associated with high levels of ACE and, thus, increased RAS activity. This is relevant as increased levels of ATII caused by elevated RAS activity promotes cancer progression by its actions on AT_1_R [152]. Other epidemiological studies have shown a protective benefit of RASis against colorectal cancer [153,154] and an overall reduced risk of cancer [155]. RASis have also been shown to improve the overall survival of patients with aggressive non-metastatic pancreatic ductal adenocarcinoma [156]. Although current data remain inconclusive, RASis appear to be broadly protective against cancer [157].

A retrospective study analyzing clinical data from 810 patients enrolled in two large multicenter studies investigating the role of two drugs targeting the RAS combined with statins in GB, shows no benefit in overall survival [158]. A recent trial on repurposing multiple drugs in combination with temozolomide, including two drugs that affect the RAS (i.e., captopril and celecoxib) for patients with GB, observed maintenance of good quality of life [159]. Captopril, an ACE inhibitor, and celecoxib, which inhibits cyclocoxygenase-2, reduce RAS activity [19]. In addition, RASis, in combination with bevacizumab, improve survival in patients with GB [160], although there is no overall survival benefit of this VEGF inhibitor as a monotherapy for de novo or recurrent GB [161]. PRR may be a critical biomarker and a therapeutic target for the treatment of GB with its connections to V-ATPase function [162], and the Wnt/β-catenin, MAPK/ERK, and PI3K/AKT/mTOR pathways [135,136,149,163] (Figure 1). Several other steps of the RAS pathway can potentially be targeted [164]. The effects of a novel approach, targeting the RAS, its bypass loops, and converging pathways simultaneously using multiple repurposed drugs on the quality of life and progression-free survival in GB patients are currently being investigated in a clinical trial [165]. Therapeutic options may be facilitated by augmenting the compensatory mechanisms of the RAS [136,164,165,166].

## 5. Recent Developments

Recent technological breakthroughs in generating human cerebral organoids [167] from pluripotent cells, combined with genetic engineering [168], mass spectroscopic proteomics [169], and next generation gene sequencing tools [170], allow more detailed investigation into the GB TME, and the role of the RAS in this *niche*. Cerebral organoids have been shown to more faithfully recapitulate the temporal and spatial aspects of the developing brain [171,172]. Vascularized cerebral organoids have been developed by utilizing ectopic expression of human ETS variant 2 in engineered ESCs to form a vascular-like network in organoids akin to endothelial cells [173]. In addition, VEGF has been used to induce blood vessel-like structures in cerebral organoids expressing markers associated with the BBB, namely, CD31 and claudin-5 [174]. In addition, human umbilical vein endothelial cells have been used to develop cerebral organoids with a well-developed tubular vascular structure. In another notable development, choroid plexus-like organoids modeled cerebrospinal fluid production with a selective barrier akin to the BBB, which may be used to model the BBTB in the GB TME [175,176,177]. Using RNA sequencing, moreover, GB cerebral organoid models have been shown to best mimic the cellular states and plasticity found in the GB TME compared to gliospheres, tumor organoids, and orthotopic patient-derived xenografts [177].

## 6. Conclusions

Despite intensive research into the biology and treatment of GB, the prognosis of patients with GB remains dismal. Understanding the heterogeneity of the tumor–host microenvironment in GB, the role of RAS and CSCs, and mapping salient interactions on a cellular level employing techniques, such as single-cell RNA sequencing, may lead to the discovery of potential therapeutic targets [178]. Cerebral and GB organoids represent an exciting yet relatively cost-effective way to delineate relevant signaling pathways within the GB TME, including the RAS, and provide models for developing and testing drug screening and therapeutic targets including RASis [179].

## Figures and Tables

**Figure 1 cancers-13-04004-f001:**
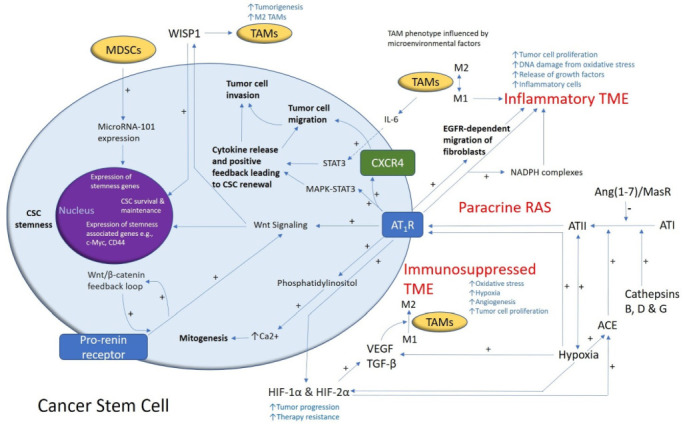
A schema demonstrating the role of the renin–angiotensin system (RAS) and its convergent signaling pathways in the glioblastoma tumor microenvironment (TME) and cancer stem cells (CSCs). A cancer stem cell (with the cytoplasm depicted in light blue and the nucleus in purple) residing within the glioblastoma TME. Angiotensin II (ATII), the physiologically active end-product of the paracrine RAS, activates ATII receptor 1 (AT_1_R) leading to increased tumor cell proliferation, oxidative stress, hypoxia and angiogenesis, and inflammation—the hallmarks of cancer. This contributes to an inflammatory TME by increasing the number of inflammatory cells, partly by increasing the number of NADPH complexes, leading to tumor cell proliferation, DNA damage from oxidative stress, and release of growth factors. AT_1_R also activates phosphatidylinositol signaling, which increases cytosolic Ca^2+^ to promote mitogenesis. Hypoxia increases paracrine RAS activity by upregulating angiotensin-converting enzyme (ACE) and the expression of hypoxia-inducible factor 1α (HIF-1α) and HIF-2α, which increase tumor progression and treatment resistance. HIF-1α, HIF-2α, and hypoxia increase the expression of vascular endothelial growth factor (VEGF) which increases angiogenesis. AT_1_R, via MAPK-STAT3 signaling, contributes to a cytokine release that leads to CSC renewal. C-X-C chemokine receptor type 4 (CXCR4) promotes tumor cell migration and invasion. AT_1_R signaling and the prorenin receptor, which act in a feedback loop with Wnt/β-catenin, increase Wnt signaling which promotes CSC stemness by upregulating stemness-associated markers. Myeloid-derived suppressor cells (MDSCs) promote CSC characteristics by increasing microRNA-101 expression that induces expression of stemness-related genes in CSCs. The Ang(1–7)/MasR axis opposes the ACE/ATII/AT_1_R axis. Cathepsins B, D, and G act as bypass loops for the RAS. Under the influence of the TME, polarization of tumor-associated macrophages (TAMs)—immune cells that are located within the TME—changes from the M1 to M2 phenotype. M2 TAMs induce the proliferation of CSCs via interleukin 6 (IL-6)-induced activation of STAT3, leading to cytokine release and positive feedback contributing to CSC renewal. Glioblastoma CSCs secrete Wnt-induced signaling protein 1 (WISP1), which facilitates a pro-tumor TME by promoting the survival of CSCs and M2 TAMs, and also promotes CSC maintenance. Abbreviations: ATI, angiotensin I; AT_2_R, ATII receptor 2; Ang(1–7), angiotensin 1–7; ATIII, angiotensin III; MAPK, mitogen-activated protein kinase. Figure modified and reproduced with permission from the *J Histochem Cytochem* [19].

**Figure 2 cancers-13-04004-f002:**
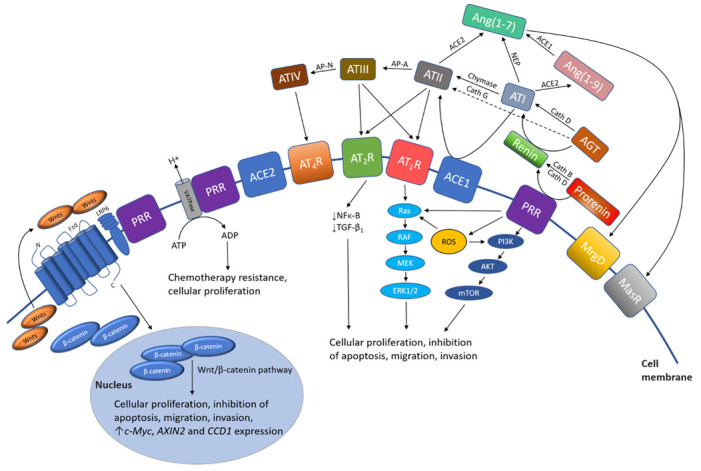
A schema showing the effect of the renin–angiotensin system (RAS) and its convergent signaling pathways on the tumor microenvironment to influence cellular proliferation, invasiveness, and cell survival in cancer development. The RAS interacts with downstream pathways, such as the Ras/RAF/MEK/ERK (light blue) pathway and the PI3K/AKT/mTOR (dark blue) pathway, and the upstream Wnt/β-catenin pathway (intermediate blue) that influence cellular proliferation, migration, inhibition of apoptosis, migration, and invasion (see text). PRR, pro-renin receptor; LRP6, low-density lipoprotein receptor-related protein; Fzd, frizzled receptor; Cath G, cathepsin G; Cath B, cathepsin B; Cath D, cathepsin D; ACE1, angiotensin-converting enzyme 1; ACE2, angiotensin-converting enzyme 2; ADP, adenosine diphosphate; AGT, angiotensinogen; ATP, adenosine triphosphate; Ang(1–7), angiotensin (1–7); Ang(1–9), angiotensin (1–9); AP-A, aminopeptidase-A; NEP, neutral endopeptidase; AP-N, aminopeptidase-N; ATI, angiotensin I; ATII, angiotensin II; ATIII, angiotensin III; ATIV, angiotensin IV; AT_1_R, angiotensin II receptor 1; AT_2_R, angiotensin II receptor 2; AT_4_R, angiotensin II receptor 4; MrgD, Mas-related-G protein coupled receptor; MasR, Mas receptor; mTOR, mammalian target of rapamycin; NF-κB, nuclear factor kappa B; TGF-β_1_, transforming growth factor-β_1_; V-ATPase, vacuolar H^+^-adenosine triphosphate.

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
