# Peer review of "The Renin–Angiotensin System in the Tumor Microenvironment of Glioblastoma"

_cancers, 2021, doi:10.3390/cancers13164004_

Round 1

Reviewer 1 Report

The authors wrote a good review that summarizes the key cellular and molecular players in the microenvironment of glioblastoma (GB) with a special emphasis on what is currently known about the contribution of the renin-angiotensin system (RAS) to GB progression. Although the implementation of the Stupp protocol (i.e., chemoradiation followed by adjuvant temozolomide) led to undeniable albeit quite modest improvements to the overall survival of GB patients, the development of more effective therapies remains an unmet medical need for these patients. Along this line, targeting various RAS components in cancer patients with widely available and low-cost drugs (e.g., ACE inhibitors, ARBs, etc.) is a tempting proposition which needs further clinical validation. Unfortunately, the current clinical data available with these compounds are far from being conclusive. Furthermore, more basic research studies are needed to further clarify the mechanistic roles played by various RAS components during the progression of GB. The authors discuss some of these caveats and suggest several future research directions. 

Author Response

We thank you for your support and insightful comments

Reviewer 2 Report

In this work, the authors exhaustively summarize the role of the Renin Angiotensin System in glioblastoma recurrence by combining two fundamental aspects, cancer stem cells and the tumor microenvironment. Furthermore, the review underscores the relevance of RAS inhibitors in improving the treatment of glioblastoma and its consequences. The review is well structured and referenced. Overall, it is good work useful for a large number of oncology researchers.

Comments:

  • In the second paragraph on page 9, the authors mention the DD genotype and its phenotypic consequences such as the accumulation of ACE. This is the only reference (at least, that I have seen) to the DD genotype in the article. Then, it should be clarified a little more. CSC homozygous for the DD genotype of the ACE 1 gene have elevated plasma ACE levels that correlate with angiotensin II levels.
  • The authors refer to the use of RAS inhibitors in a general way as RASis or drugs targeting RAS. Subsequently, captopril and celecoxib are specifically cited. It would be interesting to explain whether the role of RAS inhibition refers specifically to ACE 1 inhibitors or not. If I am not mistaken, celecoxib is a COX-2 inhibitor and ACE2 inhibitors are also used to modulate RAS signaling.
  • Cathepsin G participates in the formation of angiotensin II. It should be interesting to mention if this could be relevant in glioblastoma to bypass ACE1 inhibition.
  • In figure 2 and in the third paragraph on page 8, there is a spelling error. The cyclin D1 oncogene is CCND1.

Author Response

We thank you for your support and helpful comments. We have added a brief description to clarify our statement on elevated ACE in individuals with a DD genotype in our revised manuscript. We have also clarified that RAS inhibition refers to any RAS component being inhibited, preventing the downstream effects of its specific action. We have added a statement of the action of cathepsin G which is included in Figure 2. We trust that these amendments are now satisfactory.